# Fast Efficient Hyperparameter Tuning
# for Policy Gradient Methods

**Supratik Paul, Vitaly Kurin, Shimon Whiteson**
Deptartment of Computer Science
University of Oxford
{supratik.paul,vitaly.kurin,shimon.whiteson}@cs.ox.ac.uk

## Abstract

The performance of policy gradient methods is sensitive to hyperparameter settings that must be tuned for any new application. Widely used grid search methods for tuning hyperparameters are sample inefficient and computationally expensive. More advanced methods like Population Based Training (Jaderberg et al., 2017) that learn optimal schedules for hyperparameters instead of fixed settings can yield better results, but are also sample inefficient and computationally expensive. In this paper, we propose Hyperparameter Optimisation on the Fly (HOOF), a gradient-free algorithm that requires no more than one training run to automatically adapt the hyperparameter that affect the policy update directly through the gradient. The main idea is to use existing trajectories sampled by the policy gradient method to optimise a one-step improvement objective, yielding a sample and computationally efficient algorithm that is easy to implement. Our experimental results across multiple domains and algorithms show that using HOOF to learn these hyperparameter schedules leads to faster learning with improved performance.

## 1 Introduction

Policy gradient methods (Williams, 1992; Sutton et al., 1999) optimise reinforcement learning policies by performing gradient ascent on the policy parameters and have shown considerable success in environments characterised by large or continuous action spaces (Mordatch et al., 2015; Schulman et al., 2016; Rajeswaran et al., 2017). However, like other gradient-based optimisation methods, their performance can be sensitive to a number of key hyperparameters.

For example, the performance of first order policy gradient methods can depend critically on the learning rate, the choice of which in turn often depends on the task, the particular policy gradient method in use, and even the optimiser, e.g., RMSProp (Tieleman and Hinton, 2012) and ADAM (Kingma and Ba, 2014) have narrow ranges for good learning rates (Henderson et al., 2018b) which may not be known a priori. Even for second order methods like Natural Policy Gradients (NPG) (Kakade, 2001) or Trust Region Policy Optimisation (TRPO) (Schulman et al., 2015), which are more robust to the KL divergence constraint (which can be interpreted as a learning rate), significant performance gains can often be obtained by tuning this parameter (Duan et al., 2016).

Similarly, variance reduction techniques such as Generalised Advantage Estimators (GAE) (Schulman et al., 2016), which trade variance for bias in policy gradient estimates, introduce key hyperparameters $(\gamma, \lambda)$ that can also greatly affect performance (Schulman et al., 2016; Mahmood et al., 2018).

Given such sensitivities, there is a great need for effective methods for tuning policy gradient hyperparameters. Perhaps the most popular hyperparameter optimiser is simply grid search (Schulman et al., 2015; Mnih et al., 2016; Duan et al., 2016; Igl et al., 2018; Farquhar et al., 2018). More sophisticated techniques such as Bayesian optimisation (BO) (Srinivas et al., 2010; Hutter et al.,

2011; Snoek et al., 2012; Chen et al., 2018) have also proven effective, and new innovations such as Population Based Training (PBT) (Jaderberg et al., 2017) and meta-gradients (Xu et al., 2018) have shown considerable promise. Furthermore, a host of methods have been proposed for hyperparameter optimisation in supervised learning (see Section 4).

However, all these methods suffer from a major problem: they require performing many learning runs to identify good hyperparameters. This is particularly problematic in reinforcement learning, where it incurs not just computational costs but sample costs, as new learning runs typically require fresh interactions with the environment. This sample inefficiency is obvious in the case of grid search, BO based methods and PBT. However, even meta-gradients, which reuses samples collected by the underlying policy gradient method to train the meta-learner, requires multiple training runs. This is because the meta-learner introduces its own set of hyperparameters, e.g., meta learning rate and reference $(\gamma, \lambda)$, all of which need tuning to achieve good performance.

Furthermore, grid search and BO based methods typically estimate only the best fixed values of the hyperparameters, which often actually need to change dynamically during learning (Jaderberg et al., 2017; François-Lavet et al., 2015). This is particularly important in reinforcement learning, where the distribution of visited states, the need for exploration, and the cost of taking suboptimal actions can all vary greatly during a single learning run.

To make hyperparameter optimisation practical for reinforcement learning methods such as policy gradients, we need radically more efficient methods that can dynamically set key hyperparameters on the fly, not just find the best fixed values, and do so within a single run, using only the data that the baseline method would have gathered anyway, without introducing new hyperparameters that need tuning. This goal may seem ambitious, but in this paper we show that it is actually entirely feasible, using a surprisingly simple method we call Hyperparameter Optimisation on the Fly (HOOF).

The main idea is as follows: At each iteration, sample trajectories using the current policy. Next, generate some candidate policies and estimate their value sample efficiently by using an *off-policy* method. Finally, update the policy greedily with respect to the estimated value of the candidates. In practice, HOOF uses the policy gradient method with different hyperparameter (e.g., the learning rate, $\gamma$, and $\lambda$) settings to generate candidate policies and then uses importance sampling (IS) to construct off-policy estimates of the value of each candidate policy.

The viability of such a simple approach is counter-intuitive since off-policy evaluation using IS tends to have high variance that grows rapidly as the behaviour and evaluation policies diverge. However, HOOF is motivated by the insight that in second order methods such as NPG and TRPO, constraints on the magnitude of the update in policy space ensure that the IS estimates remain informative. While this is not the case for first order methods, we show that adding a simple KL constraint, without any of the complications of second order methods, suffices to keep IS estimates informative and enable effective hyperparameter optimisation. We further show that the performance of HOOF is robust to the setting of this KL constraint.

HOOF is 1) sample efficient, requiring no more than one training run; 2) computationally efficient compared to sequential and parallel search methods; 3) able to learn a dynamic schedule for the hyperparameters that outperforms methods that learn fixed hyperparameter settings; and 4) simple to implement. Being gradient free, HOOF also avoids the limitations of gradient-based methods (Sutton, 1992; Luketina et al., 2016; Xu et al., 2018) for learning hyperparameters. While such methods can be more sample efficient than grid search or PBT, they can be sensitive to the choice of their own hyperparameters (see Sections 4 and 5.1) and thus require more than one training run to tune their own hyperparameters.

We evaluate HOOF across a range of simulated continuous control tasks using the Mujoco OpenAI Gym environments (Brockman et al., 2016). First, we apply HOOF to A2C (Mnih et al., 2016), and show that using it to learn the learning rate can improve performance. We also perform a benchmarking exercise where we use HOOF to learn both the learning rate and the weighting for the entropy term and compare it against a grid search across these two hyperparameters. Next, we show that using HOOF to learn optimal hyperparameter schedules for NPG can outperform TRPO. This suggests that while strictly enforcing the KL constraint enables TRPO to outperform NPG, doing so becomes unnecessary once we can properly adapt NPG's hyperparameters.

## 2 Background

Consider the RL task where an agent interacts with its environment and tries to maximise its expected return. At timestep $t$, it observes the current state $s_t$, takes an action $a_t$, receives a reward $r_t = r(s_t, a_t)$, and transitions to a new state $s_{t+1}$ following some transition probability $\mathcal{P}$. The value function of the state $s_t$ is $V(s_t) = \mathbb{E}_{a \sim \pi, s \sim \mathcal{P}}[\sum_{i=0}^{\infty} \gamma^i r_{t+i}]$ for some discount rate $\gamma \in [0, 1)$. The undiscounted formulation of the objective is to find a policy that maximises the expected return $J(\pi) = \mathbb{E}_{a \sim \pi, s \sim \mathcal{P}, s_0 \sim p(s_0)}[\sum_t r_t]$. In stochastic policy gradient algorithms, $a_t$ is sampled from a parametrised stochastic policy $\pi(a|s)$ that maps states to actions. These methods perform an update of the form

$$\pi' = \pi + f(\psi). \tag{1}$$

Here $f(\psi)$ represents a step along the gradient direction for some objective function estimated from a batch of sampled trajectories $\{\tau_1^\pi, \tau_2^\pi, \dots, \tau_K^\pi\}$, and $\psi$ is the set of hyperparameters. We use $\pi$ to denote both the policy as well as the parameters.

For policy gradient methods with GAE, $\psi = (\alpha, \gamma, \lambda)$, and the update takes the form:

$$f(\alpha, \gamma, \lambda) = \alpha \underbrace{\sum_t \nabla \log \pi(a_t|s_t) A_t^{GAE(\gamma, \lambda)}}_{g(\gamma, \lambda)} \tag{2}$$

where $A_t^{GAE(\gamma, \lambda)} = (1 - \lambda)(A_t^{(1)} + \lambda A_t^{(2)} + \lambda^2 A_t^{(3)} + \dots)$ with $A_t^{(k)} = -V(s_t) + r_t + \gamma r_{t+1} + \dots + \gamma^{k-1} r_{t+k-1} + \gamma^k V(s_{t+k})$. By discounting future rewards and bootstrapping off the value function, GAE reduces the variance due to rewards observed far in the future, but adds bias to the policy gradient estimate. Well chosen $(\gamma, \lambda)$ can significantly speed up learning (Schulman et al., 2016; Henderson et al., 2018a; Mahmood et al., 2018).

In first order methods, small updates in parameter space can lead to large changes in policy space, leading to large changes in performance. Second order methods like NPG address this by restricting the change to the policy through the constraint $KL(\pi'||\pi) \leq \delta$. An approximate solution to this constrained optimisation problem leads to the update rule:

$$f(\delta, \gamma, \lambda) = \sqrt{\frac{2\delta}{g(\gamma, \lambda)^T I(\pi)^{-1} g(\gamma, \lambda)}} I(\pi)^{-1} g(\gamma, \lambda), \tag{3}$$

where $I(\pi)$ is the estimated Fisher information matrix (FIM).

Since the above is only an approximate solution, the $KL(\pi'||\pi)$ constraint can be violated in some iterations. Further, since $\delta$ is not adaptive, it might be too large for some iterations. TRPO addresses these issues by requiring an improvement in the surrogate $\mathcal{L}_\pi(\pi') = \mathbb{E}_{a \sim \pi, s \sim \mathcal{P}}[\frac{\pi'(a|s)}{\pi(a|s)} A^{GAE(\gamma, \lambda)}]$, as well as ensuring that the KL-divergence constraint is satisfied. It does this by performing a backtracking line search along the gradient direction. As a result, TRPO is more robust to the choice of $\delta$ (Schulman et al., 2015).

## 3 Hyperparameter Optimisation on the Fly

The main idea behind HOOF is to automatically adapt the hyperparameters during training by greedily maximising the value of the updated policy, i.e., starting with policy $\pi_n$ at iteration $n$, HOOF sets

$$\psi_n = \underset{\psi}{\text{argmax}} \, J(\pi_{n+1})$$
$$= \underset{\psi}{\text{argmax}} \, J(\pi_n + f(\psi)), \tag{4}$$

Given a set of sampled trajectories, $f(\psi)$ can be computed for any $\psi$, and thus we can generate different candidate $\pi_{n+1}$ without requiring any further samples. However, solving the optimisation problem in (4) requires evaluating $J(\pi_{n+1})$ for each such candidate. Any on-policy approach would have prohibitive sample requirements, so HOOF uses weighted importance sampling (WIS) to

---

**Algorithm 1** HOOF

---

**input** Initial policy $\pi_0$, number of policy iterations $N$, search space for $\psi$, KL constraint $\epsilon$ if using first order policy gradient method.
1: **for** n = 0, 1, 2, 3, ..., N **do**
2:     Sample trajectories $\tau_{1:K}$ using $\pi_n$.
3:     **for** z = 1, 2, ... Z **do**
4:         Generate candidate hyperparameter $\{\psi_z\}$ from the search space.
5:         Compute candidate policy $\pi^z$ using $\psi_z$ in (1)
6:         Estimate $J(\pi^z)$ using WIS (5)
7:         Compute $KL(\pi^z||\pi_n)$ if using first order policy gradient method,
8:     **end for**
9:     Select $\psi_n$, and hence $\pi_{n+1}$, according to (7) or (4)
10: **end for**

---

construct an off-policy estimate of $J(\pi_{n+1})$. Given sampled trajectories $\{\tau_1^{\pi_n}, \tau_2^{\pi_n}, .., \tau_K^{\pi_n}\}$, with corresponding returns $\{R_1^{\pi_n}, R_2^{\pi_n}, ..., R_K^{\pi_n}\}$, the WIS estimate of $J(\pi_{n+1})$ is given by:

$$J(\pi_{n+1}) = \sum_{k=1}^{K} \left( \frac{w_k}{\sum_{k=1}^{K} w_k} \right) R_k^{\pi_n}, \tag{5}$$

where $w_k = \frac{P(\tau_k^{\pi_n} \sim \pi_{n+1})}{P(\tau_k^{\pi_n} \sim \pi_n)}$. Since $p(\tau|\pi) = p(s_0) \prod_{i=0}^{T} \pi(a_i|s_i) p(s_{i+1}|s_i, a_i)$, the transitions cancel out and we have:

$$w_k = \frac{\prod_{i=0}^{T} \pi_{n+1}(a_i|s_i^k)}{\prod_{i=0}^{T} \pi_n(a_i|s_i^k)}. \tag{6}$$

The success of this approach depends critically on the quality of the WIS estimates, which can suffer from high variance that grows rapidly as the distributions of $\pi_{n+1}$ and $\pi_n$ diverge. Fortunately, for natural gradient methods like NPG, $KL(\pi_{n+1}||\pi_n)$ is automatically approximately bounded by the update, ensuring reasonable WIS estimates when HOOF directly uses (4). In the following, we consider the more challenging case of first order methods.

### 3.1 First Order HOOF

Without a KL bound on the policy update, it may seem that WIS will not yield adequate estimates to solve (4). However, a key insight is that, while the estimated policy value can have high variance, the relative ordering of the policies, which HOOF solves for, has much lower variance (See Appendix E for an illustrative example). Nonetheless, HOOF could still fail if $KL(\pi_{n+1}||\pi_n)$ becomes too large, which can occur in first order methods. Hence, First Order HOOF modifies (4) by constraining $KL(\pi_{n+1}||\pi_n)$:

$$\psi_n = \underset{\psi}{\operatorname{argmax}} \, J(\pi_{n+1}) \quad \text{s.t.} \quad KL(\pi_{n+1}||\pi_n) < \epsilon. \tag{7}$$

While this yields an update that superficially resembles that of natural gradient methods, the KL constraint is applied only during the search for the optimal hyperparameter settings using WIS. The direction of the update is determined solely by a first order gradient update rule, and estimation and inversion of the FIM is not required. From a practical perspective, this constraint is enforced by computing the KL for each candidate policy based on the observed trajectories, and the candidate is rejected if this sample KL is greater than the constraint.

If learning the learning rate using HOOF, we can also use the KL constraint to dynamically adjust the search bounds: At each iteration, if none of the candidates violate the KL constraint, we increase the upper bound of the search space by a factor $\nu$, while if a large proportion of the candidates violate the KL constraint, we reduce the upper bound by $\nu$. This makes HOOF even more robust to the initial setting of the search space. Note that this is entirely optional, and is simply a means to reduce the number of number of candidates that would otherwise need to be generated and evaluated to ensure that a good solution of (4) is found.

## 3.2 $(\gamma, \lambda)$ Conditioned Value Function

If we use HOOF to learn $(\gamma, \lambda)$, $g_n$ has to be computed for each setting of $(\gamma, \lambda)$. With neural net value functions, we modify our value function such that its inputs are $(s, \gamma, \lambda)$, similar to Universal Value Function Approximators (Schaul et al., 2015). Thus we learn a $(\gamma, \lambda)$-conditioned value function that can make value predictions for any candidate $(\gamma, \lambda)$ at the cost of a single forward pass. In Appendix D we present some experimental results to show that learning a $(\gamma, \lambda)$-conditioned value function is key to the success of HOOF.

## 3.3 Robustness to HOOF Hyperparameters and Computational Costs

HOOF introduces two types of hyperparameters of its own: the search spaces for the various hyperparameters it tunes, and the number of candidate policies generated for evaluation. Since the candidate policies are generated using random search, these hyperparameters express a straight up trade-off between performance and computational cost: A larger search space and larger number of candidates should lead to better solution for (4), but incur higher computational cost. However, just like in random search, the generation and evaluation of the candidate policies can be performed in parallel to reduce wall clock time. Alternatively, Bayesian Optimisation could be used to solve (4) efficiently. Finally, we note that HOOF with random search is always more computationally efficient than grid/random search over the hyperparameters with the same number of candidates, as HOOF saves on the additional computational cost of sampling trajectories for each candidate incurred by grid/random search. HOOF additionally introduces the KL constraint hyperparameter for first order methods. We show experimentally that the performance of HOOF is robust to a wide range of settings for this.

## 3.4 Choice of Optimiser

Throughout this paper we use random search as the optimiser for (4) to show that the simplest methods suffice. However, any gradient-free optimiser could be used instead. For example, grid search, CMA-ES (Hansen and Ostermeier, 2001), or Bayesian Optimisation (Brochu et al., 2010) are all viable alternatives.

Gradient based methods are not viable for two reasons. First, they require that $J(\pi_{n+1})$ be differentiable w.r.t. the hyperparameters, which might be difficult or impossible to compute, e.g. with the TRPO update. Second, they introduce learning rate and initialisation hyperparameters, which require tuning at the expense of sample efficiency.

## 4 Related Work

Most hyperparameter search methods can be broadly classified into sequential search, parallel search, and gradient based methods.

Sequential search methods perform a training run with some candidate hyperparameters, and use the results to inform the choice of the next set of hyperparameters for evaluation. BO is a sample efficient global optimisation framework that models performance as a function of the hyperparameters, and is especially suited for sequential search as each training run is expensive. After each training run BO uses the observed performance to update the model in a Bayesian way, which then informs the choice of the next set of hyperparameters for evaluation. Several modifications have been suggested to further reduce the number of evaluations required: input warping (Snoek et al., 2014) to address nonstationary fitness landscapes; freeze-thaw BO (Swersky et al., 2014) to decide whether a new training run should be started and the current one discontinued based on interim performance; transferring knowledge about hyperparameters across similar tasks (Swersky et al., 2013); and modelling training time as a function of dataset size (Klein et al., 2016). To further speed up the wall clock time, some BO based methods use a hybrid mode wherein batches of hyperparameter settings are evaluated in parallel (Contal et al., 2013; Desautels et al., 2014; Shah and Ghahramani, 2015; Wang et al., 2016; Kandasamy et al., 2018).

By contrast, parallel search methods like grid search and random search run multiple training runs with different hyperparameter settings in parallel to reduce wall clock time, but require more parallel computational resources. These methods are easy to implement, and have been shown to perform well (Bergstra et al., 2011; Bergstra and Bengio, 2012).

Both sequential and parallel search suffer from two key disadvantages. First, they require performing multiple training runs to identify good hyperparameters. Not only is this computationally inefficient, but when applied to RL, also sample inefficient as each run requires fresh interactions with the environment. Second, these methods learn fixed values for the hyperparameters that are used throughout training instead of a schedule, which can lead to suboptimal performance (Luketina et al., 2016; Jaderberg et al., 2017; Xu et al., 2018).

PBT (Jaderberg et al., 2017) is a hybrid of random and sequential search, with the added benefit of adapting hyperparameters during training. It starts by training a population of hyperparameters which are then updated periodically to further explore promising hyperparameter settings. However, by requiring multiple training runs, it inherits the sample inefficiency of random search.

HOOF is much more sample efficient because it requires no more interactions with the environment than those gathered by the underlying policy gradient method for one training run. Consequently, it is also far more computationally efficient. However, while HOOF can only optimise hyperparameters that directly affect the policy update, these methods can tune other hyperparameters, e.g., policy architecture. Combining these complementary strengths in an interesting topic for future work.

Gradient based methods (Sutton, 1992; Bengio, 2000; Luketina et al., 2016; Pedregosa, 2016; Xu et al., 2018) adapt the hyperparameters by performing gradient descent on the policy gradient update function with respect to the hyperparameters. This raises the fundamental problem that the update function needs to be differentiable. For example, the update function for TRPO uses conjugate gradient to approximate $I(\pi)^{-1}g$, performs a backtracking line search to enforce the KL constraint, and introduces a surrogate improvement constraint, which introduce discontinuities in the update and makes it non-differentiable.

A second major disadvantage of these methods is that they introduce their own set of hyperparameters, which can make them sample inefficient if they require tuning. For example, the meta-gradient estimates can have high variance, which in turn significantly affects performance. To address this, the objective function of meta-gradients introduces reference $(\gamma', \lambda')$ hyperparameters to trade off bias and variance. As a result, its performance can be sensitive to these, as the experimental results of Xu et al. (2018) show. Furthermore, gradient based methods tend to be highly sensitive to the setting of the learning rate, and these methods introduce their own learning rate hyperparameter for the meta learner which requires tuning, as we show in our experiments. As a gradient-free method, HOOF does not require a differentiable objective and, while it introduces a few hyperparameters of its own, these do not affect sample efficiency, as mentioned in Section 3.3.

Other work on non-gradient based methods includes that of Kearns and Singh (2000), who derive a theoretical schedule for the TD($\lambda$) hyperparameter that they show is better than any fixed value. Downey et al. (2010) learn a schedule for TD($\lambda$) using a Bayesian approach. White and White (2016) greedily adapt the TD($\lambda$) hyperparameter as a function of state. Unlike HOOF, these methods can only be applied to TD($\lambda$) and, in the case of Kearns and Singh (2000), are not compatible with function approximation.

# 5 Experiments

To experimentally validate HOOF, we apply it to four simulated continuous control tasks from MuJoCo OpenAI Gym (Brockman et al., 2016): HalfCheetah, Hopper, Ant, and Walker. We start with A2C, and show that HOOF performs better than multiple baselines, and is also far more sample efficient. Next, we use NPG as the underlying policy gradient method and apply HOOF to learn $(\delta, \gamma, \lambda)$ and show that it outperforms TRPO.

We repeat all experiments across 10 random starts. In all figures solid lines represent the median, and shaded regions the quartiles. Similarly all results in tables represent the median. Hyperparameters that are not tuned are held constant across HOOF and baselines to ensure comparability. Details about all hyperparameters can be found in the appendices, and code is available at `https://github.com/supratikp/HOOF`.

## 5.1 HOOF with A2C

In the A2C framework, a neural net with parameters $\theta$ is commonly used to represent both the policy and the value function, usually with some shared layers. The update function (1) for A2C is a linear

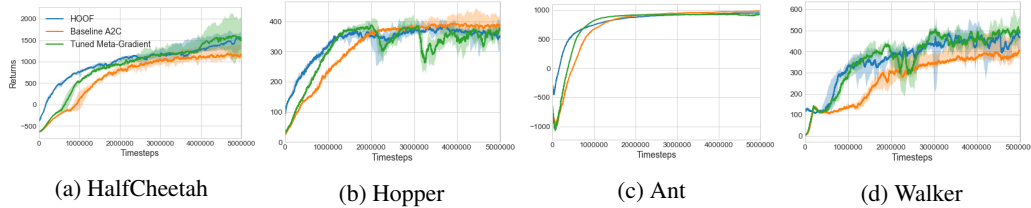

| (a) HalfCheetah | (b) Hopper | (c) Ant | (d) Walker |

Figure 1: Performance of HOOF with $\epsilon = 0.03$ compared to Baseline A2C and Tuned Meta-Gradients. The hyperparameters $(\alpha_0, \beta)$ of meta gradients had to be tuned using grid search which required 36x the samples used by HOOF.

Table 1: Performance of HOOF with different values of the KL constraint ($\epsilon$ parameter). The results show that the performance is relatively robust to the setting of $\epsilon$.

| KL constraint | $\epsilon = 0.01$ | $\epsilon = 0.02$ | $\epsilon = 0.03$ | $\epsilon = 0.04$ | $\epsilon = 0.05$ | $\epsilon = 0.06$ | $\epsilon = 0.07$ |
|---|---|---|---|---|---|---|---|
| HalfCheetah | 1,203 | 1,451 | 1,524 | 1,325 | 1,388 | 1301 | 1504 |
| Hopper | 359 | 358 | 350 | 362 | 359 | 370 | 365 |
| Ant | 916 | 942 | 952 | 957 | 971 | 963 | 969 |
| Walker | 466 | 415 | 467 | 475 | 456 | 402 | 457 |

combination of the gradients of the policy loss, the value loss, and the policy entropy:

$$f_\theta(\alpha) = \alpha\{\nabla_\theta \log \pi_\theta(a|s)(R - V_\theta(s)) + c_1 \nabla_\theta(R - V_\theta(s))^2 + c_2 \nabla_\theta H(\pi_\theta(s))\}, \qquad (8)$$

where we have omitted the dependence on the timestep and other hyperparameters for ease of notation. The performance of A2C is particularly sensitive to the choice of the learning rate $\alpha$ (Henderson et al., 2018b), which requires careful tuning.

We learn $\alpha$ using HOOF with the KL constraint $\epsilon = 0.03$ ('HOOF'). We compare this against two baselines: (1) Baseline A2C, i.e., A2C with the initial learning rate set to the OpenAI Baselines default (0.0007), and (2) learning rate being learnt by meta-gradients ('Tuned Meta-Gradient'), where the hyperparameters introduced by meta-gradients were tuned using grid search.

The learning curves in Figure 1 shows that across all environments HOOF learns faster than Baseline A2C, and also outperforms it in HalfCheetah and Walker, demonstrating that learning the learning rate online can yield significant gains.

The update rule for meta-gradients when learning $\alpha$ reduces to $\alpha' = \alpha + \beta \nabla_{\theta'} \log \pi_{\theta'}(a|s)(R - V_{\theta'}(s))\frac{f_\theta(\psi)}{\alpha}$, where $\beta$ is the meta learning rate. This leads to two issues: what should the learning rate be initialised to ($\alpha_0$), and what should the meta learning rate be set to? Like all gradient based methods, the performance of meta gradients can be sensitive to the choices of these two hyperparamters. When we set $\alpha_0$ to the OpenAI baselines default setting and $\beta$ to 0.001 as per Xu et al. (2018), A2C fails to learn at all. Thus, we had to run a grid search over $(\alpha_0, \beta)$ to find the optimal settings across these hyperparameters. In Figure 1 we plot the best run from this grid search. Despite using 36 times as many samples (due to the grid search), meta-gradients still cannot outperform HOOF, and learns slower in 3 of the 4 tasks. The returns for each of the 36 points on the grid are presented in Appendix B.1 and they show that the performance of meta gradients can be sensitive to these two hyperparamters.

To show that HOOF's performance is robust to $\epsilon$, its own hyperparameter quantifying the KL constraint, we repeated our experiments with different values of $\epsilon$. The results presented in Table 1 show that HOOF's performance is stable across different values of this parameter. This is not surprising – the sole purpose of the constraint is to ensure that the WIS estimates remain viable.

Finally, to ascertain the sample efficiency of HOOF relative to grid search, we perform a benchmarking exercise. We used HOOF to learn both the learning rate and the entropy coefficient ($c_2$ in (8)). We split the search bounds for these across a grid with 11x11 points and ran A2C for each setting on the grid. For computational reasons we set the budget for each training run to 1 million timesteps. Given a budget of $n$ training runs, we randomly subsample $n$ points from the grid (without replacement) and note the best return. We repeat this 1000 times to get an estimate of the expected best return

Table 2: Comparison of sample efficiency of HOOF over grid search.

| | HOOF Returns | Max return over subsampled grid of size | | | |
| | | 1 | 2 | 5 | 10 |
|---|---|---|---|---|---|
| HalfCheetah | 702 | -558 | -241 | 113 | 354 |
| Hopper | 321 | 109 | 165 | 240 | 287 |
| Ant | 675 | -7561 | -272 | 177 | 476 |
| Walker | 175 | 99 | 153 | 224 | 279 |

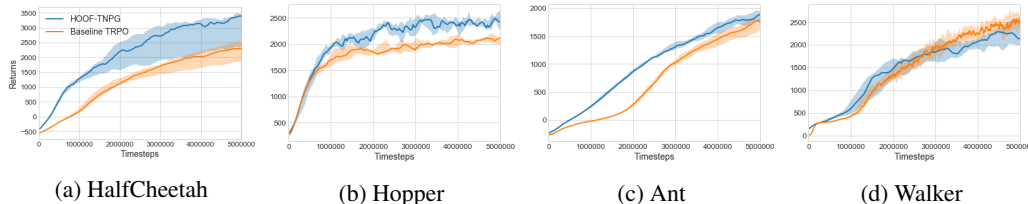

(a) HalfCheetah  (b) Hopper  (c) Ant  (d) Walker

Figure 2: Performance of HOOF-TNPG vs TRPO baselines.

of the grid search with a budget of $n$ training runs. The results presented in Table 2 compares the returns of HOOF to that of the expected best return for grid search with different training budgets. The performance of grid search is much worse than that of HOOF with the same budget (i.e., only 1 training run). The results show that grid search can take more than 10 times as many samples to match HOOF's performance.

Appendix A.3 contains further experimental details, including results confirming that the KL constraint is crucial to ensuring sound WIS estimates.

In Appendix A.4 we show that HOOF is also robust to the choice of the optimiser by running the experiments with SGD (instead of RMSProp) as the optimiser. In this case the difference in performance is highly significant with Baseline A2C failing to learn at all.

## 5.2 HOOF with Truncated Natural Policy Gradients (TNPG)

A major disadvantage of natural policy gradient methods is that they require the inversion of the FIM in (3), which can be prohibitively expensive for large neural net policies with thousands of parameters. TNPG (Duan et al., 2016) and TRPO address this by using the conjugate gradient algorithm to efficiently approximate $I(\pi)^{-1}g$. TRPO has been shown to perform better than TNPG in continuous control tasks (Schulman et al., 2015), a result attributed to stricter enforcement of the KL constraint.

However, in this section, we show that stricter enforcement of the KL constraint becomes unnecessary once we properly adapt TNPG's learning rate. To do so, we apply HOOF to learn $(\delta, \gamma, \lambda)$ of TNPG ('HOOF-TNPG'), and compare it with TRPO with the OpenAI Baselines default settings of $(\epsilon = 0.01, \gamma = 0.99, \lambda = 0.98)$ ('Baseline TRPO').

Figure 2 shows the learning curves of HOOF-TNPG and the Baseline TRPO. HOOF-TNPG learns much faster, and outperforms Baseline TRPO in all environments except for Walker where there's no significant difference. Figure 3 presents the learnt $(\delta, \gamma, \lambda)$. The results show that different KL constraints and GAE hyperparameters are needed for different domains. We could not compare with meta-gradients as the objective function is not differentiable, as discussed earlier in Section 4. We also could not perform a comparison against grid search similar to the one in Section 5.1 as the computational burden of performing a grid search over three hyperparameters was too large.

## 6 Conclusions & Future Work

The performance of a policy gradient method is highly dependent on its hyperparameters. However, methods typically used to tune these hyperparameters are highly sample inefficient, computationally expensive, and learn only a fixed setting of the hyperparameters. In this paper we presented HOOF, a sample efficient method that automatically learns a schedule for the learning rate and GAE hyperparameters of policy gradient methods without requiring multiple training runs. We believe that this,

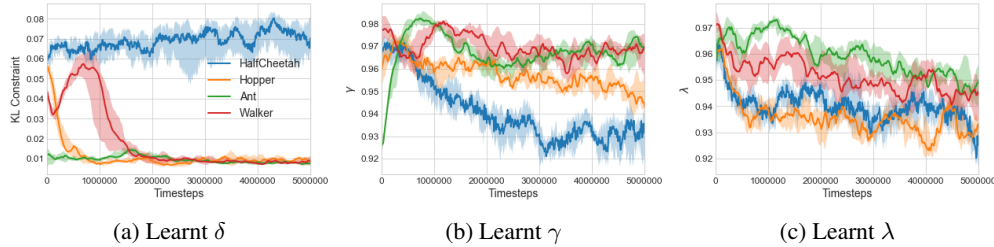

|  | (a) Learnt $\delta$ | (b) Learnt $\gamma$ | (c) Learnt $\lambda$ |

Figure 3: Hyperparameters learnt by HOOF-TNPG for HalfCheetah, Hopper, and Walker.

combined with its simplicity and ease of implementation, makes HOOF a compelling method for optimising policy gradient hyperparameters.

While we have presented HOOF as a method to learn the hyperparameters of a policy gradient algorithm, the underlying principles are far more general. For example, one could compute a distribution for the gradient and generate candidate policies by sampling from that distribution, instead of just using the point estimate of the gradient. It has also been hypothesised that state/action dependent discount factors might help speed up learning (White, 2017; Fedus et al., 2019). This could be achieved by using HOOF to learn the parameters of a function that maps the states/actions to the discount factors.

### Acknowledgements

This project has received funding from the European Research Council (ERC) under the European Union's Horizon 2020 research and innovation programme (grant agreement #637713), and Samsung R&D Institute UK. The experiments were made possible by a generous equipment grant from NVIDIA.

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
