[Supplementary Material · hoof_supp.pdf]

# A  A2C Experimental Details

We present further details about our A2C experiments in this section.

## A.1  Implementation Details

Our codebase for the A2C experiments is based on OpenAI Baselines (Dhariwal et al., 2017) implementation of A2C and uses their default hyperparameters. Experiments involving HOOF use the same hyperparameters apart from those that are learnt by HOOF. All hyperparameters are presented in Table 3.

Table 3: Hyperparameters used for A2C experiments.

| Hyperparameter | Value |
| --- | --- |
| Number of environments (num_envs) | 40 |
| Timesteps per worker (nsteps) | 5 |
| Total environment steps | 5e6 |
| Discounting $\gamma$ | 0.99 |
| Max gradient norm | 0.5 |
| Optimiser | RMSProp |
| $-\alpha$ | 0.99 |
| $-\epsilon$ | 1e-5 |
| Policy | MLP |
| – Number of fully connected layers | 2 |
| – Number of units per layer | 64 |
| – Activation | tanh |
| Default settings for Baseline A2C (and HOOF and meta gradients if not learnt) | |
| – Initial learning rate | 7e-4 |
| – Learning rate schedule | linear annealing |
| – Value function cost weight ($c_1$) | 0.5 |
| – Entropy cost weight ($c_2$) | 0.01 |
| Grid search over $(\alpha, c_2)$ with A2C | |
| – Grid settings for $\alpha$ | $0.01 \times 10^{-0.5 \times \{0,1,..,10\}}$ |
| – Grid settings for $c_2$ | $0.05 \times \{0.0, 0.1, .., 1.0\}$ |
| HOOF specific hyperparameters | |
| – Initial search bounds for $\alpha$ | [0,1e-2] |
| – Number of random samples for $\alpha$ | 100 |
| – Search bounds for $c_2$ (for grid search experiment) | [0,0.2] |
| – Number of random samples for $c_2$ (for grid search experiment) | 50 |

For HOOF, $\alpha_{UB}$, the upper bound of the search space for $\alpha$ was dynamically updated at each iteration based on the following: if no candidates violate the KL constraint, $\alpha_{UB} \leftarrow 1.25\alpha_{UB}$. If more than 80% of the candidates violate the KL constraint, $\alpha_{UB} \leftarrow \alpha_{UB}/1.25$.

## A.2  Learnt Hyperparameters

The hyperparameters learnt by HOOF are presented in Figure 4.

## A.3  Performance of HOOF without a KL Constraint

Figure 5 shows that without a KL constraint HOOF does not converge, which confirms that we need to constrain policy updates so that WIS estimates remain sound.

## A.4  Robustness to the Choice of Optimiser

OpenAI implementation of A2C uses RMSProp as the default optimiser. To check how robust HOOF's performance is to the choice of the optimiser, we ran both Baseline A2C and HOOF with

Figure 4: Schedule for the learning rates learnt by HOOF. Refer to Equation (8).

(a) HalfCheetah  (b) Hopper  (c) Ant  (d) Walker2d

Figure 5: Comparison of the performance of HOOF with $\epsilon = 0.03$ and HOOF without any KL constraint.

SGD instead. The learning curves presented in Figure 6 shows that in this case HOOF's performance is far better than that of Baseline A2C which fails to learn at all.

## B  Meta-Gradients Update for $\alpha$

The meta-gradient algorithm for hyperparameters $\psi$ proceeds as follows: 1) Sample trajectories $\tau_{1:K}^\theta \sim \pi_\theta$. 2) Update $\theta' = \theta + f_\theta(\psi)$ (where $f_\theta$ is as per (8)). 3) Sample trajectories $\tau_{1:K}^{\theta'} \sim \pi_{\theta'}$. 4) Update $\psi' = \psi + \beta \frac{\partial J'(\tau_{1:K}^{\theta'}, \bar\psi)}{\partial \theta'} \frac{\partial f_\theta(\psi)}{\partial \psi}$, where $J'$ is the meta-objective with $\bar\psi$ the set of reference hyperparameters introduced by the meta-gradient algorithm to balance bias-variance tradeoff within the meta-objective, and $\beta$ is the meta learning rate. For $\psi = \alpha$, $\frac{\partial f_\theta(\psi)}{\partial \psi} = \frac{f_\theta(\psi)}{\alpha}$, and we can use the policy loss as the meta objective, with $\frac{\partial J'(\tau_{1:K}^{\theta'}, \bar\psi)}{\partial \theta'} = \nabla_{\theta'} \log \pi_{\theta'}(a|s)(R - V_{\theta'}(s))$.

An unconstrained meta-update can lead to $\alpha$ being negative. Clipping $\alpha$ to 0 after each meta update is not feasible since it leads to the situation where the policy does not update at all. Hence a log transform was used instead to ensure $\alpha > 0$.

### B.1  Results of Grid Search for Meta-Gradients

The returns after 5 millions timesteps for each setting of $(\alpha_0, \beta)$ on the grid is given in Table 4. Note that very few settings of the hyperparameters can match the performance of HOOF, while some settings of $(\alpha_0, \beta)$ can lead to the algorithm failing to learn at all. Setting $\alpha_0 = 1e-3$, which is closest to the OpenAI Baselines default setting, and $\beta = 1e-3$ as was used by Xu et al. (2018) in their experiments leads to performance well below that of HOOF, or even Baseline A2C.

It is also important to note that HOOF only requires 1 training run of samples (i.e. 5 million timesteps) while the grid search over the hyperparameters means that meta-gradients requires 36x samples to be able to match HOOF.

## C  TNPG Experimental Details

We present further details about our TNPG experiments in this section.

(a) HalfCheetah     (b) Hopper     (c) Ant     (d) Walker2d

Figure 6: Comparison of the performance of HOOF with $\epsilon = 0.03$ and Baseline A2C where the optimiser is SGD for both (instead of RMSProp).

(a) HalfCheetah     (b) Hopper     (c) Ant     (d) Walker

Figure 7: Performance of HOOF without $(\gamma, \lambda)$ conditioned value functions: Not learning $(\gamma, \lambda)$ conditioned value functions leads to significant reduction in performance in all environments except Ant.

### C.1 Implementation Details

Our codebase for the TNPG experiments is based on OpenAI Baselines (Dhariwal et al., 2017) implementation of TRPO and uses their default hyperparameters. Experiments involving HOOF use the same hyperparameters apart from those that are learnt by HOOF. All hyperparameters are presented in Table 5.

## D    Importance of Learning $(\gamma, \lambda)$ Conditional Value Functions

In Figure 7 we compare the performance of HOOF ('HOOF-TNPG') with that of HOOF without $(\gamma, \lambda)$ conditional value functions ('HOOF-no-$(\gamma, \lambda)$'). Clearly the conditioning is key to good performance. This is because the value is highly dependent on $(\gamma, \lambda)$ which changes throughout training.

## E    Illustration of the Ordering Effect of WIS

We illustrate the assertion about the relative ordering of WIS estimates through a simple experiment: Let $p(x) = N(0, 1)$ be our behaviour distribution. We are interested in $E_{q_i(x)}[X^2]$ where $q_i(x) = N(\mu_i, 1)$, $\mu_i = \{0, 1, 2, 3, 4, 5\}$. We can compute the true value analytically as $1 + \mu_i^2$. Now we compare this to a WIS estimate: we sample 10 points from $p(x)$ and use them to estimate $E_{q_i(x)}[X^2]$. We repeat this 1000 times. The boxplot of the WIS estimates in Fig 8a shows that we cannot rely on them directly as they becomes worse as $q_i(x)$ diverges from $p(x)$. However, in Fig 8b we see that the relative ordering is reliable. Note this does not guarantee that by using WIS to estimate the value of each candidate policy will always lead to a correct solution in (4). However, any factor that leads to better estimates of the policy value (for example, increasing the number of trajectories sampled) is also likely to lead to a better estimate of the relative ordering which (4) relies on.

Table 4: Results of grid search over meta-gradients hyperparameters. * denotes algorithm diverged with returns $< -10^5$. Settings with returns greater than HOOF returns are shown in bold.

**HalfCheetah**

| $\alpha_0$ | | $\beta$ 1e-2 | 1e-3 | 1e-4 | 1e-5 | 1e-6 | 1e-7 | HOOF |
|---|---|---|---|---|---|---|---|---|
| | 1e-2 | -10,972 | -1,230 | * | * | * | -498 | |
| | 1e-3 | -7,137 | -221 | 468 | 1,080 | **1,568** | 1,272 | |
| | 1e-4 | * | -245 | 313 | 441 | -223 | 86 | 1523 |
| | 1e-5 | * | -247 | 324 | -499 | -515 | -518 | |
| | 1e-6 | -641 | -224 | -404 | -618 | -616 | -631 | |
| | 1e-7 | -643 | -351 | -611 | -633 | -638 | -639 | |

**Hopper**

| $\alpha_0$ | | $\beta$ 1e-2 | 1e-3 | 1e-4 | 1e-5 | 1e-6 | 1e-7 | HOOF |
|---|---|---|---|---|---|---|---|---|
| | 1e-2 | -2,950 | -13,271 | -808 | -1,845 | 87 | 103 | |
| | 1e-3 | -12,045 | -508 | -801 | 54 | **378** | **368** | |
| | 1e-4 | -20,086 | 68 | 67 | 225 | 215 | 236 | 350 |
| | 1e-5 | -3,309 | 70 | 65 | 67 | 61 | 61 | |
| | 1e-6 | 35 | 67 | 63 | 64 | 64 | 50 | |
| | 1e-7 | -7,793 | 72 | 64 | 54 | 20 | 18 | |

**Ant**

| $\alpha_0$ | | $\beta$ 1e-2 | 1e-3 | 1e-4 | 1e-5 | 1e-6 | 1e-7 | HOOF |
|---|---|---|---|---|---|---|---|---|
| | 1e-2 | * | 393 | * | * | * | * | |
| | 1e-3 | * | 761 | 752 | 950 | 926 | 884 | |
| | 1e-4 | -47,393 | -156 | 687 | 672 | 666 | 655 | 952 |
| | 1e-5 | * | 375 | 588 | -595 | -739 | -692 | |
| | 1e-6 | * | 283 | 373 | -1,081 | -1,073 | -1,006 | |
| | 1e-7 | -1,257 | -514 | -361 | -1,017 | -1,042 | -964 | |

**Walker**

| $\alpha_0$ | | $\beta$ 1e-2 | 1e-3 | 1e-4 | 1e-5 | 1e-6 | 1e-7 | HOOF |
|---|---|---|---|---|---|---|---|---|
| | 1e-2 | -10,316 | -2,922 | 109 | 294 | 176 | 159 | |
| | 1e-3 | -1,383 | -2,636 | 28 | 445 | **492** | **485** | |
| | 1e-4 | -931 | -153 | 31 | 220 | 133 | 124 | 467 |
| | 1e-5 | -4,732 | -117 | 125 | 112 | 116 | 121 | |
| | 1e-6 | -47,222 | -3,005 | 137 | 113 | 39 | 12 | |
| | 1e-7 | -774 | -22 | 111 | 113 | 2 | 0 | |

Figure 8: In (a) the WIS estimates of $E_{q_i(x)}[X^2]$ diverges from the true values as $q_i(x)$ diverges from $p(x)$. However (b) shows that the relative ordering based on the WIS estimates is reliable.

Table 5: Hyperparameters used for TNPG experiments.

| | |
|---|---|
| Total environment steps | 5e6 |
| Timesteps per iteration | 10,000 |
| Policy | MLP |
| – Number of fully connected layers | 2 |
| – Number of units per layer | 64 |
| – Activation | tanh |
| Baseline TRPO | |
| – KL constraint | 0.01 |
| – Discounting $\gamma$ | 0.99 |
| – GAE-$\lambda$ | 0.98 |
| HOOF specific hyperparameters | |
| – Search bounds for $\epsilon$ | [0.001, 0.1] |
| – Search bounds for ($\gamma$) | [0.85, 1] |
| – Search bounds for ($\lambda$) | [0.85, 1] |
| – Number of random samples for $\epsilon$ | 50 |
| – Number of random samples for ($\gamma, \lambda$) | 50 |