[Reviews · NeurIPS 2019]

Reviewer 1



I agree that the problem is very important and I like the line of thinking on this. I am ultimately lukewarm overall because of questions I have about how to characterize the idea and whether/how it is really better: I'm concerned about the point I already alluded to. The algorithm is really just to perform zeroth order optimization over policies (eq 4) using weighted importance sampling to evaluate (eq 5). The "hyperparameter tuning" is only an artifact of the method used to generate candidates for that optimization. The main contribution of the algorithm is that it does the optimization using eqs 4/5 in a way that (claims to) introduce no hyperparameters that require substantial tuning. That is useful if its true, but it begs the question: are there other ways to optimize eq 4? Maybe you can use gradient descent on it directly? If you do, does it start to look like other methods? It thus seems like the key point is somewhere in lines 141-172. There are new hps introduced in place of the ones that are replaced but the argument is that the new ones tend not to need tuning. There are many reasons why this could be the case but its unclear to me whether/why it really is. Maybe its just that zeroth order optimization over a small number of candidate policies is less prone to overfitting the high-variance weighted importance sampling evaluation than a method that would more thoroughly optimize it. The caption in figure 1 implies that the curves for the competing algorithms need to be shifted by 36X. Is this true? If so, perhaps they should be plotted this way. It seems the range of epsilons in table 1 should be larger, perhaps on a log scale. The fact that even the 1st derivatives of the results across each row are not monotonic suggests that the results are dominated by noise. Lines 291-294. You compared to baseline TRPO but not the tuned version. I understand it takes many many more samples to tune TRPOs parameters but it seems possible that having done that, TRPO would still beat HOOF-TNPG with those many many more samples. Smaller points: I think you never actually defined g. Its relatively obvious from its appearance in eq 2 but should still be defined. Figure 1 is too small to read.

Reviewer 2



Summary: This paper tackles the problem of automatic hyperparameter tuning. The proposed approach differs notably from prior approaches in being more sample efficient, requiring data collected from only a single policy. After collecting a batch of data, candidate policies are generated by sampling a set of hyperparameters and taking a gradient step on the current policy with them. The value of these candidates is evaluated by importance weighting the batch of data collected by the original policy, and the best is chosen as the new policy. Originality: I am not familiar with all the related work, but the idea seems novel. Quality: As noted in the paper, the approach only works for tuning parameters that only affect the policy update (for example, you cannot tune the batch size with HOOF). It seems like the most sensitive parameters are often these (things like learning rate or discount), so this limitation does not seem too severe. This is more of a sanity check for hyperparameter optimization for PG algorithms in general, but I would like to understand if it’s important to find “good” hyperparameters, or simply to change them a lot over the course of training. Can you reference a study of this? For example, what if you took the median in Eq. 4, or even a random update? One drawback here seems to be the random sampling method for hyperparameter options (Section 3.3) - this search space will grow exponentially as the number of hyperparameters increases. The method really hinges on the importance sampling estimates being reasonable. Can you elaborate on the statement that “while the estimated policy value can have high variance, the relative ordering of the policies has much lower variance.” Why is this necessarily true? If the values are high variance, then they could easily be out of order? The results seem significant to me, though I’m not familiar with the state-of-the-art in this area. Can HOOF be used to find hyperparameter schedules that can be used to train another model - first of all on the same environment, and then even on a different one? Clarity: The paper is mostly well-written and clear. From my understanding, the main contribution of this paper is its sample rather than computational efficiency, and that should be highlighted. Running PBT with on-policy algorithms on a real system would be totally infeasible, but HOOF would be feasible. Maybe this is what is meant by “within a single training run”, but I think this language can be made much more clear. If you define “training run” as training a population in parallel, then it’s also true of PBT, which is not what you mean. I think it is misleading to refer to the algorithm as “meta-learning” in the abstract - is this a typo? It occurs nowhere else in the paper. I suggest shortening the introduction considerably - a lot of it is redundant with the related work section. In Figure 1, it would be good to provide some results from runs with suboptimal hyperparameters, to get a sense of the variance in returns from different settings. In Figure 2, it would be good to show TNPG without HOOF, to demonstrate it is indeed worse than TRPO. Significance: medium - seems like a reasonable and sample efficient approach -------------- Post-rebuttal ------------------ Thanks for humoring the sanity check with random sampling rather than zero-order optimizing Eq. 4. As a small comment, I think explicitly referring to the method as “zero-order optimization” would be quite helpful in making the method clear at first read. Thanks also for your graphical demonstration of the WIS estimates ordering. My point was more that it I think it is *possible* for them to be out of order, even though I agree they will certainly be in order more often than individual value estimates will be. And this depends on K, does it not? If you only sampled 1 trajectory, you’d just have one estimate per candidate (I know that’s unrealistic for PG algorithms, but just for example).

Reviewer 3



The paper presented a method called HOOF for automatic hyperparameter tuning of policy gradient methods that is computationally and sample efficient. The basic idea is to perform greedy optimization of the hyperparameters at each step such that updating the policy for one step using those hyperparameters leads to the best policy. At each timestep, different policy candidates are generated by updating the current policy network using different hyperparameters. Then, these candidates are ranked according to weighted importance sample (WIS) estimates of the cumulative reward using trajectories collected by the current policy, and the best update is chosen. Also, in case of first-order methods where the action distributions can change drastically after each update, the KL divergence of the action distributions of the policy after update is bounded so that the WIS estimates are reliable. The proposed method is used for hyperparameter tuning of A2C and NPG in four MuJoCo benchmarks against manually tuned hyperparameters and meta-gradients. It is shown that the proposed method is able to achieve similar or slightly better performance on these benchmarks by finding the right hyperparameters within a single run. The paper is fairly well-written and easy to follow. Previous works on automating hyperparameter selection require multiple sequential or parallel runs, which causes a drastic reduction in sample-efficiency. It is very interesting to see developments on tuning hyperparameters within a single run. However, the work presented in the paper is limited to policy gradient methods and it cannot be used to tune the architectures of the networks. I hope that it inspires future works along the same direction for policy-based and also value-based and model-based RL algorithms. Comments: - Why is \pi used to denote both the policy function as well as it's parameters? - What is the effect of learning (\gamma, \lambda) conditional value functions in the performance reported in Figure 2? Is it just about computational efficiency or does it also have an effect on the performance of the agent? - In Table 1, the sensitivity of \epsilon is measured in the range of values [0.01, 0.07]. How much does \epsilon have to change to see significant change in the performance? Post-rebuttal comments: Thanks for the clarification. I recommend acceptance.

[Author Response · NeurIPS 2019]



(a) HalfCheetah

(b) Hopper

(c) HalfCheetah

(d) Hopper

Figure 1: (a) and (b) compares the performance of HOOF-A2C with different settings of the KL constraint ($\epsilon$). Clearly $\epsilon = 0.001$ is quite conservative with slow learning while $\epsilon = 1.0$ is too aggressive. In line with existing methods that rely on KL constraints (like TRPO/NPG), we believe that [0.01, 0.1] is a reasonable range and as we have demonstrated, HOOF is robust to settings within this range. In (c) and (d) we compare performance of HOOF-TNPG with two ablations: HOOF-Random where $(\gamma, \lambda)$ is chosen randomly (instead of $\text{argmax}$ in Eq 4), and HOOF-no-$(\gamma, \lambda)$ where the value function does not condition on $(\gamma, \lambda)$. Clearly both of these are key to good performance. The performance of the latter is similar to that of HOOF-Random since the value function predictions are quite meaningless, leading to updates that are essentially random. We could not present results for Ant and Walker due to space constraints.

**Reviewer 1**: The ultimate goal of any hyperparameter optimisation method is to remove the need for expensive manual
tuning of the hyperparameters. Our experiments demonstrate that HOOF achieves this goal and we believe this makes it
a hyperparameter optimisation algorithm. The fact that it uses a zeroth order optimiser to perform a fresh hyperparameter
search at each iteration of the policy gradient algorithm does not detract from its usefulness in achieving this goal.

There a couple of issues with using gradient based methods to solve Eq 4: 1) This requires that $J(\pi_{n+1})$ be differentiable
wrt the hyperparameters, which might be difficult to compute or impossible, e.g. with the TRPO update, and 2) it
introduces a learning rate and initialisation hyperparameter, which will require tuning thereby sacrificing sample
efficiency. Thus we are restricted to zero order optimisers. We used random search to show that the simplest methods
can still work well, however we could use any zero order optimiser like Bayesian Optimisation/CMA-ES.

For natural gradients like TNPG, HOOF does not add any new hyperparameters beyond those used by grid search - i.e.
the range of the search space, and the number of points in the grid, and these simply express a tradeoff between compute
and performance. Larger ranges and finer grids require more compute, but are likely to result in better performance, and
the same applies to HOOF. Other methods like PBT introduce more hyperparameters than these.

For first order methods, and only if learning the learning rate, HOOF additionally adds the KL constraint hyperparameter
epsilon. We disagree that a log scale evaluation of epsilon is warranted – unlike learning rate, a search over the KL
constraint (for methods like NPG/TRPO) is usually done on a linear scale. That said, we have presented the results for
$\epsilon = \{0.001, 0.01, 0.03, 0.1, 1.0\}$ in Figs 1a and 1b, together with comments in the caption.

Fig 1 caption is correct, and is to show that even after taking 36x samples meta-gradients can't do better than HOOF.

It is possible that highly tuned TRPO might outperform HOOF, but at a cost of an order of magnitude more samples. If
we had a budget for that many samples, instead of using WIS to estimate $J(\pi_{n+1})$ in Eq 4, we could do an on-policy
evaluation and then there is no a priori reason to believe that HOOF would underperform tuned TRPO, since the noise
in solution of Eq 4 due to WIS estimates goes away.

**Reviewer 2**: Refer to Figs 1c and 1d for comparison to the
suggested random baseline. We agree that random search does
not scale well with dimensionality of $\psi$ – we could use CMA-
ES or other gradient-free optimisers that scale better instead.

We demonstrate the point about relative ordering of WIS es-
timates empirically. Let $p(x) = N(0,1)$ be our behaviour
distribution. We are interested in $E_{q_i(x)}[X^2]$ where $q_i(x) =$
$N(\mu_i, 1)$, $\mu_i = \{0, 1, 2, 3, 4, 5\}$. We can compute the true
value analytically as $1 + \mu_i^2$. Now we compare this to a WIS
estimate: we sample 10 points from $p(x)$ and use them to es-
timate $E_{q_i(x)}[X^2]$. We repeat this 1000 times. The boxplot of
the WIS estimates in Fig 2a shows that we cannot rely on them
directly as they becomes worse as $q_i(x)$ diverges from $p(x)$.
However, in Fig 2b we see that the relative ordering is reliable.

(a)

(b)

Figure 2: In (a) the WIS estimates of $E_{q_i(x)}[X^2]$ diverges from the true values as $q_i(x)$ diverges from $p(x)$. However (b) shows that the relative ordering based on the WIS estimates is reliable.

**Reviewer 3**: Please refer to Figs 1c and 1d for the effect of not learning a value function conditioned on $(\gamma, \lambda)$
('HOOF-no-$(\gamma, \lambda)$'), and Figs 1a and 1b for the effect of the KL-constraint, together with some comments.

[Meta-Review · NeurIPS 2019]

The paper introduces a new method for hyperparameter optimization in policy gradient, "HOOF" (hyperparameter optimization on the fly). In the reviews and the discussions, we saw that all reviewers appreciated the method's sample efficiency, computational efficiency and performance vs. simple baselines. All three reviews recommended accept: R2 gave the strongest endorsement, R3 gave a borderline accept, and while R1 also gave a score of 7, the tone of the review was a bit equivocating, with statements such as "I am ultimately lukewarm overall because of questions I have about how to characterize the idea and whether/how it is really better."